# The Effect of Intermittent versus Continuous Non-Invasive Blood Pressure Monitoring on the Detection of Intraoperative Hypotension, a Sub-Study

**DOI:** 10.3390/jcm11144083

**Published:** 2022-07-14

**Authors:** Marije Wijnberge, Björn van der Ster, Alexander P. J. Vlaar, Markus W. Hollmann, Bart F. Geerts, Denise P. Veelo

**Affiliations:** 1Department of Anesthesiology, Amsterdam UMC—Location AMC, University of Amsterdam, Meibergdreef 9, 1105 AZ Amsterdam, The Netherlands; j.p.stervander@amsterdamumc.nl (B.v.d.S.); m.w.hollmann@amsterdamumc.nl (M.W.H.); d.p.veelo@amsterdamumc.nl (D.P.V.); 2Department of Intensive Care, Amsterdam UMC—Location AMC, University of Amsterdam, Meibergdreef 9, 1105 AZ Amsterdam, The Netherlands; 3Healthplus.ai, 1013 CN Amsterdam, The Netherlands; b.f.geerts@hotmail.com

**Keywords:** hemodynamics, perioperative, anesthesiology, surgery

## Abstract

Intraoperative hypotension is associated with postoperative complications. However, in the majority of surgical patients, blood pressure (BP) is measured intermittently with a non-invasive cuff around the upper arm (NIBP-arm). We hypothesized that NIBP-arm, compared with a non-invasive continuous alternative, would result in missed events and in delayed recognition of hypotensive events. This was a sub-study of a previously published cohort study in adult patients undergoing surgery. The detection of hypotension (mean arterial pressure below 65 mmHg) was compared using two non-invasive methods; intermittent oscillometric NIBP-arm versus continuous NIBP measured with a finger cuff (cNIBP-finger) (Nexfin, Edwards Lifesciences). cNIBP-finger was used as the reference standard. Out of 350 patients, 268 patients (77%) had one or more hypotensive events during surgery. Out of the 286 patients, 72 (27%) had one or more missed hypotensive events. The majority of hypotensive events (92%) were detected with NIBP-arm, but were recognized at a median of 1.2 (0.6–2.2) minutes later. Intermittent BP monitoring resulted in missed hypotensive events and the hypotensive events that were detected were recognized with a delay. This study highlights the advantage of continuous monitoring. Future studies are needed to understand the effect on patient outcomes.

## 1. Introduction

An association between intraoperative hypotension and postoperative renal insufficiency, myocardial injury and increased mortality in non-cardiac surgical patients has been reported in numerous cohort studies [1,2,3,4]. Randomized clinical trials showed that maintaining an optimal blood pressure (BP) during surgery reduced the risk of postoperative organ dysfunction [5,6]. In 2019, the Perioperative Quality Initiative consensus statement concluded that anesthesiologists should aim to maintain a mean arterial pressure (MAP) above 60–70 mmHg during surgery [7].

Intraoperatively, BP can be monitored continuously or intermittently. The current standard for continuous BP monitoring is invasively via cannulation of the radial artery. Placement of an arterial cannula poses a small risk of developing nerve damage, infection, thrombus formation or a pseudoaneurysm [8,9,10]. A finger BP cuff employing volume clamp technology allows for non-invasive continuous measurement of BP during surgery (cNIBP-finger) [11]. MAP values measured by cNIBP-finger have shown to be comparable to invasive arterial BP [12,13,14].

In the vast majority of surgical patients, however, BP is monitored intermittently using an oscillometric method with a non-invasive cuff around the upper arm (NIBP-arm) [15]. On average, NIBP-arm is measured every 2–5 min which could potentially lead to a delay in recognition or missed hypotensive events. As intraoperative hypotension occurs frequently and even short durations of intraoperative hypotension may be harmful, wider implementation of continuous monitoring could be of benefit [2,16,17]. A recent randomized controlled trial has shown that continuous versus intermittent monitoring halved the time-weighted average (TWA) of intraoperative hypotension [18]. That study compared two non-invasive BP monitoring techniques, similar to the present study. No studies have yet assessed the delay time between recognition with NIBP-arm versus cNIBP-finger.

Our primary objective was to determine whether use of intermittent (NIBP-arm) compared with continuous (cNIBP-finger) BP monitoring results in missed hypotensive events. This is not a validation study; we purely studied the effect of continuous monitoring. Our second objective was to assess the delay time between continuous and intermittent BP monitoring in the recognition of hypotensive events. We hypothesize that intermittent BP monitoring would result in missed hypotensive events and would result in delayed recognition of hypotensive events. In an exploratory manner, we assessed the effect of NIBP-arm sample interval on the number of missed events and delay time.

## 2. Materials and Methods

The present study describes a sub-study from a prior published prospective cohort study [15]. The study is written according to the Strobe guidelines for cohort studies [19,20]. The local medical ethical committee of the Amsterdam University Medical Centers (UMC), location AMC, provided a waiver for the study (W15_080#15.0094, 11 March 2015). The trial was registered at clinicaltrials.gov with registration number NCT03533205. Data were collected in two phases, between April and October 2015 and between May and December 2016. Adult patients (>18 years of age) undergoing surgery were included. During surgery, BP was monitored as per standard care and additionally with cNIBP-finger. Standard care could entail either invasive BP monitoring with cannulation of the radial artery or with oscillometric NIBP-arm monitoring. Subjects were excluded when technical problems or strong local vasoconstriction (i.e., cold fingers) prevented cNIBP-finger measurements.

For this sub-study, those patients receiving NIBP-arm as standard care (opposed to invasive arterial BP monitoring) and experiencing at least one hypotensive event during surgery were selected (see Appendix A).

### 2.1. Study Measurements

Prior to induction, a cNIBP finger cuff (Nexfin, Edwards Lifesciences Corp., Irvine, CA, USA) was connected to the patient and the heart reference sensor was zeroed at heart level. The Nexfin measured non-invasive finger BP continuously using the volume clamp method. The cuff pressure varied dynamically to keep the volume of the finger arteries under the cuff constant throughout the cardiac cycle [11,21]. The finger BP was reconstructed based on the brachial BP waveform using a physiological transfer function developed employing a large clinical database [22,23]. Care givers were blinded to the Nexfin monitor in order to prevent guidance of clinical practice based on those data.

NIBP-arm was measured with a BP cuff around the upper arm (Comfort Check™ Long, Salter Labs, Arvin, CA, USA). The NIBP-arm cuff was inflated intermittently and the interval was chosen by the treating anesthesiologist. cNIBP-finger was connected contralateral from NIBP-arm to allow continuous monitoring. Per institutional practice, BP was treated when MAP dropped below 65 mmHg.

### 2.2. Data Collection

cNIBP-finger data were extracted from the Nexfin device and NIBP-arm data were extracted from the electronical medical records system (EPIC version 2016, EPIC Systems Corporation, Verona, WI, USA and Metavision 5.46.38, iMDsoft, Tel Aviv, Israel). Patient data were collected and de-identified.

### 2.3. Sample Size

No sample size analysis was performed as data for the present sub-study were derived from an earlier published prospective cohort study and no inferential statistics were performed [15]. The results from this sub-study analysis are presented using descriptive statistics only.

### 2.4. Data Analysis

For the analysis of this study, cNIBP-finger arterial waveform data after the start of surgery (surgical incision) were included. Nexfin samples blood pressure at 200 Hz. Data was extracted from the device after internal online beat-detection was completed. Values for MAP were averaged for every 20 s. Data points during a period of poor or noisy signal quality were excluded from further analyses [15].

Hypotension was defined as a cNIBP-finger MAP below 65 mmHg for at least one minute [7]. cNIBP-finger MAP was used as reference standard. cNIBP-finger-determined hypotension was presented as total number of hypotensive events, number of hypotensive events per patient, absolute time spent in hypotension, percentage of time spent in hypotension during surgery, the area under the threshold and the time-weighted average in hypotension. The TWA of hypotension is measured by calculating the area under the threshold (AUT) divided by the total duration of surgery: time-weighted average = (depth of hypotension in millimeters of mercury below a MAP of 65 mmHg × time in minutes spent below a MAP of 65 mmHg)/total duration of the operation in minutes [24,25].

NIBP-arm intermittent data points were interpolated to allow time synchronization between cNIBP-finger and NIBP-arm (Appendix A). All patients were visually checked for time synchronization, independently by two authors (BS and MW).

Primary endpoint: missed hypotensive events were calculated as cNIBP-finger hypotensive events (MAP below 65 mmHg for more than one minute) not recognized by NIBP-arm. A missed hypotensive event based on >5 mmHg offset between cNIBP-finger and NIBP-arm was not counted as a true missed event.

Because NIBP-arm provides intermittent data, one NIBP-arm data point of a MAP below 65 mmHg was sufficient to count as a recognized event. For the missed hypotensive events, the lowest cNIBP-finger MAP value reached and the average cNIBP-finger MAP for the hypotensive events were reported. The average cNIBP-finger MAP was calculated by adding all blood pressure values during the hypotensive event divided by the number of data points. For example, for a hypotensive event with MAP 62, 60, 56, 54, 60, 64 mmHg, the average MAP would be 59 mmHg (all values/6) and the lowest MAP would be 54 mmHg.

Secondary endpoint: the time from recognition of a hypotensive event with cNIBP-finger to recognition with NIBP-arm was presented as the delay in detection time.

Exploratively, the missed events and delay times per NIBP-arm sample interval subgroup were reported. The NIBP-arm sample interval was the sample interval (e.g., 1, 2, 3, 4, 5 or more minutes) most frequently chosen during surgery. As subgroups had different numbers of patients, the missed events per subgroup had to be corrected to allow for comparison. We presented the number of missed hypotensive events as a percentage of the number of patients per subgroup.

Data analyses were performed with MATLAB and SPSS. Continuous data were presented as median with interquartile range (IQR), or mean with standard deviation (SD) when normally distributed. Categorical data were given as frequencies with percentages.

## 3. Results

### 3.1. Study Population

In the database consisting of 507 patients, a median of 2.3% of the data [IQR 0.6–9.7], which had poor signal quality, was removed. For this sub-study, 404 out of 507 patients receiving NIBP-arm as standard care (and the blinded cNIBP-finger monitoring for study purposes) were selected [15]. The other 103 excluded patients were monitored employing invasive blood pressure monitoring. Out of those 404 patients, 54 had unavailable electronical medical records for NIBP-arm data. Out of the remaining 350 patients, 268 patients (77%) had at least one hypotensive event (cNIBP-finger) during surgery and were included in our analyses. In 24 patients, missed events were based on an offset of >5 mmHg between cNIPB-finger and NIBP-arm and those events were not counted as true missed events.

The median age was 56 years (IQR 43–66) and 54% of the patients were female. The study group was heterogenous in terms of types of surgeries. The majority of anesthesiologists set the NIBP-arm interval at 3 min (43%), followed by 2 min (28%) and 5 min (20%) (see Table 1 and Appendix A).

### 3.2. Primary Endpoint

In 268 patients, 1006 total hypotensive events were recognized with cNIBP-finger, whereas 80 (8%) of these events were missed by NIBP-arm (see Table 2). The 80 missed events were distributed over 72 patients; in other words, 72 out of the 286 patients (27%) had one or more hypotensive event(s). Sixty-five patients had one missed hypotensive event, six patients had two missed hypotensive event and one patient had three missed hypotensive events (see Appendix A). The median lowest MAP for the missed hypotensive events was 59.7 mmHg (IQR 57.0–61.4) and the median average MAP for the missed hypotensive events was 61.9 mmHg (IQR 60.2–63.0).

### 3.3. Secondary Endpoint

The median delay time between the cNIBP-finger and NIBP-arm was 1.2 min (0.6–2.2).

### 3.4. Exploratory Analyses

The fraction of missed events, corrected for the number of patients per group, did increase with increasing sample intervals up to five minutes, but paradoxically showed a decrease at a sample interval of five minutes or higher (see Table 3). The delay times increased slightly as the NIBP-arm sample interval increased (see Figure 1). To illustrate, in patients with a NIBP-arm sample interval of two minutes, NIBP-arm detected hypotension a median of 1.0 min (0.5–2.3) later compared with cNIBP-finger. In patients with a NIBP-arm sample interval of five minutes, the median delay time was 1.4 min (0.9–2.5).

## 4. Discussion

Intraoperatively, intermittent BP monitoring resulted in one or more missed hypotensive events in 27% of the patients. The majority of hypotensive events were detected with intermittent BP monitoring; however, hypotensive events were recognized with a median delay time exceeding one minute. The majority of anesthesiologists measure NIBP-arm every two, three or five minutes. Notably, it is not common to measure NIBP-arm every four minutes.

As expected, the delay time between recognition of a hypotensive event increased when the sample interval increased. Paradoxically, more missed hypotensive events were recognized in patients where NIBP-arm was measured every three minutes compared with those with a five-minute sample interval. Selection bias might be the underlying cause of this finding, as for hemodynamically more stable patients the measurement interval is more often set at 5 min. Moreover, the small subgroups in these exploratory analyses might also explain this observation.

This study adds to previous work demonstrating that continuous BP monitoring reduces intraoperative hypotension [18,26,27]. However, our work is different from previous studies as we report missed events and delay time. The hypotensive events that were detected by cNIBP-finger but missed with NIBP-arm occurred between two NIBP-arm measurements. The lowest median MAP during these missed events was 60 mmHg, which is not considered a very important drop in blood pressure. It makes sense that the missed hypotensive events that resolved before the next NIBP-arm measurement do not represent severe hypotensive events. More severe hypotensive events would present as a delay in detection time as the underlying pathophysiological cause leading to the hypotension would still be present and the hypotension would not resolve spontaneously. However, a brief moment of hypotension between two NIBP-arm measurements could easily be iatrogenic, for example caused by a short drop in venous return because of compression by the surgeon. In addition, it is possible that treatments were administered between two NIBP-arm measurements. Because of the nature of this cohort study, the causes of the short missed hypotensive events remain speculative. In the majority of cases, hypotension was recognized with intermittent monitoring (NIBP-arm), but with a median delay time of 1.2 min. This is an important outcome. Earlier recognition with continuous monitoring enables earlier treatment. One could argue that missing one minute of hypotension is of limited clinical relevance; however, patients often experience more than one episode of hypotension intraoperatively, and thus delay times add up. In the present study we demonstrated a median of three (IQR 2–5) hypotensive events per patient. Additionally, previous studies have suggested even short periods of hypotension to be hazardous [2].

The continuous non-invasive device we used in this study has substantial costs. It requires an extra monitor system in the operating room and, contrary to NIBP-arm, a new finger BP cuff is required for every patient [28]. A study based on Monte Carlo simulations concludes that prevention of hypotension in a hospital with an annual volume of 10.000 non-cardiac surgical patients is associated with mean cost reductions ranging from 1.2 to 4.6 million American dollars per year. The authors calculated that the estimated mean marginal cost reduction per surgical patient linked to acute kidney injury (AKI) and myocardial injury after non-cardiac surgery (MINS) was around 272 dollars [29]. The costs of non-invasive continuous BP monitoring devices are variable. Not all patients develop intraoperative hypotension. In our study sample, 268 out of 350 patients (77%) had at least one hypotensive event. In these patients, non-invasive continuous monitoring resulted in more hypotensive events being recognized, and earlier detection of these events. Future studies should assess the cost-effectiveness of continuous non-invasive BP monitoring.

This study has some limitations. First, to answer our study question we had to use two different methods to measure BP: the current oscillometric standard (NIBP-arm) and a continuous alternative (NIBP-finger) which utilizes the arterial pressure waveform [21,30]. Although previous studies have demonstrated that—at the group level—cNIBP-finger can be interchangeably used as an alternative for NIBP-arm [12,13,14], at the patient level, an offset of >5 mmHg (the validity criterion proposed by the Association for the Advancement of Medical Instrumentation) between the two devices can exist [31]. We predefined the study to exclude those offset events. Including these events would have resulted in additional missed events.

Second, cNIBP-finger was connected contralaterally from NIBP-arm to allow continuous monitoring. A previous study demonstrated no relevant cNIBP measurement differences between contralateral side measurements [32].

Third, the 8% missed events in the present study were in between two intermittent BP measurements. As this was not an intervention trial, we do not know if these hypotensive events resolved with or without treatment. The median minimal missed MAP during those missed episodes was 60 mmHg, which is generally considered mild hypotension. However, evidence that intraoperative hypotension is hazardous is increasing and not detecting those hypotensive events in patients could lead to a false sense of safety [33].

Fourth, we took the cNIBP-finger as the reference standard in this study, and we calculated hypotension endpoints (such as TWA) for the continuous BP monitoring only. Since NIBP-arm measurements are intermittent we were not able to reliably calculate a true TWA for NIBP-arm. An additional disadvantage of the NIBP-arm is that is does not provide an arterial waveform and thus does not allow for pulse wave analysis. Pulse wave analysis can provide hemodynamic variables such as cardiac output (CO) and stroke volume variation (SVV) to assess the underlying cause of hypotension [25].

Fifth, in this study a hypotensive event was defined as a MAP below 65 mmHg for more than one minute in line with our previous studies [15,24,25]. This is important to keep in mind, as the definition of hypotension determines the number of missed events. For example, if one would define hypotension as any data point below a MAP of 65, the number of missed events would be substantially higher.

Six, the current study is a sub-study of a previously published paper [15]. As such, no sample size analysis and no inferential statistics were performed. The results from this sub-study analysis are presented using descriptive statistics only.

Seven, although the literature regarding the hazardousness of intraoperative hypotension is increasing, evidence is mostly based on associations reported in cohort studies. Randomized clinical trials demonstrating a causal effect between hypotension and worse postoperative outcome are sparse [5,6]. Future trials should aim to assess the impact of prevention of hypotension on postoperative outcomes.

## 5. Conclusions

In this single-center intraoperative cohort study, intermittent BP monitoring resulted in one or more missed events in 72 out of 268 patients. The majority of hypotensive events (92%) were detected with intermittent BP monitoring but were recognized at a median of 1.2 min later. As even short durations of hypotension could be hazardous, continuous monitoring might be preferred. Future studies are needed to determine the effect of continuous BP monitoring on patient outcomes and to assess cost-efficiency.

## Figures and Tables

**Figure 1 jcm-11-04083-f001:**
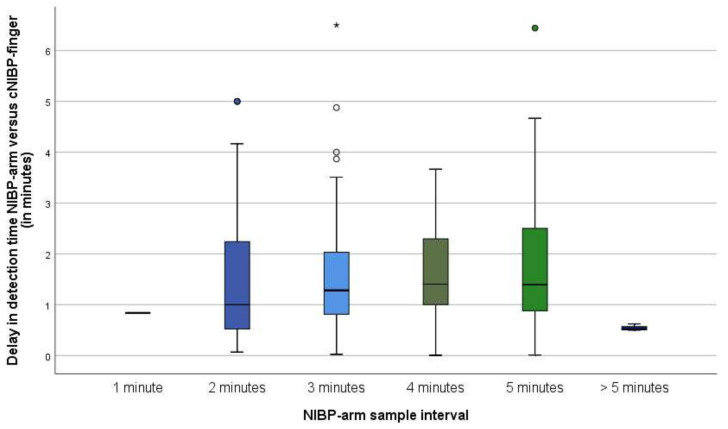
Boxplots demonstrating median delay time per NIBP-arm sample interval. The NIBP-arm sample interval was the sample interval the patient experienced for the majority of surgical time. To illustrate, if a patient had a duration of surgery of 120 min and 10 min were sampled at an interval of 2 min and the remaining 110 min were sampled at an interval of 3 min, we listed this as a sample interval of 3 min. The round dots represent outliers within the presented scale. The asterisk represents an outlier outside of the presented scale; it represents a 9.7 min delay in detection time.

**Table 1 jcm-11-04083-t001:** Baseline data of included patients.

Characteristics	*n* = 268 Patients
Age	56.0 (43.3–66.0)
Male	123 (46%)
Female	145 (54%)
Height (in cm)	173.0 (166.3–181.0)
Weight (in kg)	75.0 (65.0–88.8)
BMI	24.8 (22.6–27.8)
ASA	
*I*	105 (39.2%)
*II*	126 (47.0%)
*III*	37 (13.8%)
*IV*	0 (0%)
Length of data-collection (in hours)	2.2 (1.4–3.2)
Type of surgery:	
*Gynecological*	46 (17.2%)
*Abdominal*	50 (18.7%)
*Urological*	34 (12.7%)
*Vascular*	12 (4.5%)
*Pulmonary*	2 (0.7%)
*Trauma and orthopedic*	18 (6.8%)
*Ophthalmic*	44 (16.4%)
*Ear, nose, and throat*	37 (13.8%)
*Oral and maxillofacial*	10 (3.7%)
*Plastic*	9 (3.4%)
*Neuro*	6 (2.2%)
NIBP-arm interval (in minutes)	
*1*	1 (0.4%)
*2*	75 (28.0%)
*3*	114 (42.5%)
*4*	18 (6.7%)
*5*	54 (20.1%)
*>5 min*	6 (2.3%)

Categorical data are presented as counts with percentage. Continuous data are presented as median with interquartile range. Length of data-collection is calculated as measurement duration of cNIBP-finger. BMI = body mass index; ASA = American Society of Anesthesiologists.

**Table 2 jcm-11-04083-t002:** Hypotensive events detected with cNIBP-finger versus NIBP-arm.

	*n* = 268 Patients
Total hypotensive events ^a^	1006
Number of hypotensive events per patient ^a^	3 (IQR 2–5)
Time in hypotension ^a^ (minutes)	13.5 (4.8–31.25)
% time during surgery in hypotension ^a^	11.6 (4.1–27.4)
AUC hypotension ^a^	81.9 (28.2–205.6)
TWA hypotension ^a^	0.6 (0.2–1.6)
Total number of missed hypotensive events, NIBP-arm versus cNIBP-finger ^b^	80 (8%)
Average BP for missed events ^c^ (mmHg)	61.9 (60.2–63.0)
Lowest missed BP ^c^ (mmHg)	59.7 (57.0–61.4)
Delay in detection time (minutes), NIBP-arm versus cNIBP-finger ^d^	1.2 (0.6–2.2)

^a^ Continuous blood pressure monitoring was used as the reference standard (cNIBP-finger). ^b^ Missed hypotensive events were calculated as events detected by cNIBP-finger but not detected by intermittent NIBP-arm monitoring. ^c^ The number of patients with one of more missed hypotensive events was 72. ^d^ The delay in detection time was calculated from the onset of hypotension detected by cNIBP-finger to the first detection of the hypotensive events with NIBP-arm. BP = blood pressure.

**Table 3 jcm-11-04083-t003:** Exploratory analyses. Missed events and delay times per NIBP-arm subgroup.

	Median Delay Time (In Minutes)	Number of Patients	Total Number of Missed Events	% Missed
1 min	- *	1	0	0%
2 min	1.0 (0.5–2.3)	75	13	17%
3 min	1.3 (0.8–2.0)	114	42	36%
4 min	1.4 (0.9–2.3)	18	7	39%
5 min	1.4 (0.9–2.5)	54	17	32%
>5 min	- *	6	1	17%

* Not calculated due to small sample sizes. Min = minutes.

## Data Availability

Study data is anonymized and securely stored at the Amsterdam UMC.

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
