# Peer review of "The Effect of Intermittent versus Continuous Non-Invasive Blood Pressure Monitoring on the Detection of Intraoperative Hypotension, a Sub-Study"

_jcm, 2022, doi:10.3390/jcm11144083_

Round 1

Reviewer 1 Report

Marije Wijnberge et al. reported the results of a comparison of NIBP-arm and cNIBP-finger for detecting intraoperative hypotension in patients undergoing surgery. This demonstrated the importance of continuous blood pressure monitoring.

The results are interesting and may provide important information for further study. However, there are several concerns that must be addressed, as listed below.

1) On line 71 of P2, the total number of eligible patients is shown as 568. On the other hand, the result shows 268. Which is the correct number?

2) Several techniques were listed in Table 1. I think that Each technique had different criteria for blood pressure control. Is there any bias in the NIBP-arm interval for each technique?

Also, the percentages listed on p. 4, line 156 do not match the values in Table 1.

3) Unify the units (minutes and seconds) in Table 3. and Figure 1.

4The author should indicate the difference between the symbols (●, ○ and ☆) in Figure 1.

5) In the discussion and conclusions, the authors discuss the need to examine the postoperative impact of prevention of intraoperative hypotension. The 72 patients who developed intraoperative hypotension in this study should also be considered.

Reviewer 2 Report

This study addresses a very important, previously studied but never resolved yet issue: "Should all or most anesthetized patients have continuous monitoring of blood pressure?".

The present study is well-conceived and executed. 

However, there are a number of concerns:

1. Chief among them is the fact that the definition of hypotension is purely  numerical and lacks any correlation with patient-oriented outcomes. Thus, the lowest recorded MAP during a "hypotensive" episode was only 60mmHg and the mean 62mmHg. These are usually not very important drops in blood pressure, especially if, as in most patients, it was brief.

2. Secondly, the continuous BP measurement was performed by a long-known method, which has not really become standard and is well-known to be affected by peripheral perfusion. Using it as a gold standard is at best problematic.  

3. In addition, the authors correcly mention the logistic and economic prices to be paid for implementing this method specifically.

4. In spite of the above issues, which invalidate any potential conclusions regarding patient-centered outcomes such as acute renal failure, multiorgan failure, stroke or MI, or death, the study is worthwhile in providing information regarding the likelihood of intermittent sphyngomanometric BP measurements missing episodes of hypotension. 

5. There is a clear need to redo the study with a better-validated continuous BP measurement system, e.g. arterial line, and using patient-centered criteria such as more significant hypotension in major surgery, postoperative complications, etc. 

Reviewer 3 Report

I would like to thank the authors for their work. The purpose of this article is clearly stated. The transition between sections follows an appropriate pattern. I would like to add several comments.

Methods: I suggest that the authors list the different primary and secondary variables in a more orderly and clear way.

Discussion: Delete "The main findings of this sub-study of an earlier published prospective cohort study are:"

Thank you

Reviewer 4 Report

This study aimed to investigate whether continuous NIBP measured with a finger cuff could detect intraoperative hypotension better compared to intermittent NIBP with a brachial cuff. I read the manuscript with great interest.

I have some concerns.

At what Hz did you download the data for cNIBP data analysis? How many data points per sec or min in cNIBP?

In table 1, the range of BMI presented as 22.6 – 22.8, is it right?

The patients’ peripheral circulation or edema or body position can affect the accuracy of cNIBP measurement with volume clamp method.

Did you check the number of patients taking hypertensive agents or patients with diabetes?

I recommend presenting the patients’ position. Did you record the patients’ position (e.g. gynecological surgery- lithotomy with head down, pulmonary- lateral decubitus)?

In case of lateral decubitus position, the blood pressure would be different between the dependent and non-dependent arms.

Hypothermia can cause vasoconstriction and affect peripheral circulation; did you save the temperature data?

It is questionable whether a difference of 1.5 minutes in detecting hypotension will have a clinical impact. However, if this study shows that the delay time can be prolonged in patients with uncontrolled blood pressure or elderly patients, it will be the basis for clinical application of cNIBP. (Although this study included patients up to age 66)

Did you administer any drugs (e.g. ephedrine, phenylephrine, dopamine..) to treat the hypotension during the study periods?

I think the authors should check the paragraph breaks in the discussion section. 
